# *Lycium barbarum* Polysaccharides and Capsaicin Inhibit Oxidative Stress, Inflammatory Responses, and Pain Signaling in Rats with Dextran Sulfate Sodium-Induced Colitis

**DOI:** 10.3390/ijms23052423

**Published:** 2022-02-22

**Authors:** Yu-Shan Chen, Yu Zhi Lian, Wen-Chao Chen, Chun-Chao Chang, Alexey A. Tinkov, Anatoly V. Skalny, Jane C.-J. Chao

**Affiliations:** 1School of Nutrition and Health Sciences, Taipei Medical University, 250 Wu-Hsing Street, Taipei 110301, Taiwan; kk26652@gmail.com (Y.-S.C.); lian6239@hotmail.com (Y.Z.L.); 2Department of Dietetics, Taipei Medical University Hospital, 252 Wu-Hsing Street, Taipei 110301, Taiwan; 3Division of Gastroenterology and Hepatology, Department of Internal Medicine, Taipei Medical University Hospital, 252 Wu-Hsing Street, Taipei 110301, Taiwan; 143007@h.tmu.edu.tw (W.-C.C.); chunchao@tmu.edu.tw (C.-C.C.); 4Division of Gastroenterology and Hepatology, Department of Internal Medicine, School of Medicine, Taipei Medical University, 250 Wu-Hsing Street, Taipei 110301, Taiwan; 5Laboratory of Molecular Dietetics, I.M. Sechenov First Moscow State Medical University, 2–4 Bolshaya Pirogovskaya Street, 119435 Moscow, Russia; tinkov.a.a@gmail.com (A.A.T.); skalny3@microelements.ru (A.V.S.); 6Institute of Bioelementology, Orenburg State University, Pobedy Avenue, 13, 460018 Orenburg, Russia; 7Federal Research Centre of Biological Systems and Agrotechnologies of the Russian Academy of Sciences, 9 Yanvarya Street, 29, 460000 Orenburg, Russia; 8Master Program in Global Health and Development, Taipei Medical University, 250 Wu-Hsing Street, Taipei 110301, Taiwan; 9Nutrition Research Center, Taipei Medical University Hospital, 252 Wu-Hsing Street, Taipei 110301, Taiwan

**Keywords:** *Lycium barbarum* polysaccharides, capsaicin, oxidative stress, inflammation, pain signaling, dextran sulfate sodium, ulcerative colitis, rats

## Abstract

Ulcerative colitis (UC) is an inflammatory disease with chronic relapsing symptoms. This study investigated the effects of *Lycium barbarum* polysaccharides (LBP) and capsaicin (CAP) in dextran sulfate sodium (DSS)-induced UC rats. Rats were divided into normal, DSS-induced UC, and UC treated with 100 mg LBP/kg bw, 12 mg CAP/kg bw, or 50 mg LBP/kg bw and 6 mg CAP/kg bw. Rats were fed LBP or CAP orally by gavage for 4 weeks, and UC model was established by feeding 5% DSS in drinking water for 6 days during week 3. Oral CAP and mixture significantly reduced disease activity index. Oral LBP significantly decreased serum malondialdehyde, interleukin (IL)-6, colonic tumor necrosis factor (TNF)-α levels, and protein expression of transient receptor potential cation channel V1 (TRPV1) and transient receptor potential ankyrin 1 (TRPA1), but increased serum catalase activity. Oral CAP significantly suppressed serum IL-6, colonic TRPV1 and TRPA1 protein expression, but elevated IL-10 levels, serum superoxide dismutase and catalase activities. The mixture of LBP and CAP significantly reduced serum IL-6, colonic TNF-α and TRPA1 protein. In conclusion, administration of LBP and/or CAP attenuate DSS-induced UC symptoms through inhibiting oxidative stress, proinflammatory cytokines, and protein expression of TRPV1 and TRPA1.

## 1. Introduction

Ulcerative colitis (UC) is an inflammatory bowel disease sub-type, commonly occurring in the sigmoid colon, characterized by relapsing and remitting inflammatory mucosa [1]. The incidence and prevalence of UC have been increased worldwide, and from 2001 to 2015 the incidence and prevalence of UC in Taiwan were found to be raised from 0.54 to 0.95 per 100,000 and from 2.10 to 2.18 per 100,000 [2], respectively. The pathogenesis of UC remains unclear, but recent studies have suggested that diet, environmental factors, lifestyle, and genetic factors were crucial in its development [3,4,5].

Massive infiltration of neutrophils in the colon was found during the onset of UC disease, and following overproduction of reactive oxygen species (ROS) could contribute to the damage to the intestinal mucosa, which could exacerbate the symptoms of colitis by affecting the membrane integrity of the colonic mucosa and further leading to mucosal disorders [6,7]. Up to 70% of patients experienced abdominal pain in the onset or exacerbations of UC [8]. Inflammatory cytokines were believed to be able to stimulate primary neurons and increase the nociceptive input to neurons, resulting in aberrant signal propagation and processing [9]. This aberrant signal was referred to as visceral hypersensitivity and could explain the inflammatory pain [9].

Several drugs, such as anti-inflammatory drugs, immuno-suppressors, and antibiotics were given to UC patients for symptoms control [10], but may cause adverse effects including but not limited to weight loss, headache, and diarrhea [11]. *Lycium barbarum* L, also known as goji berry or wolfberry, was commonly used in Chinese medicine and contained bioactive substances such as polysaccharides, carotenoids, and betaine [12]. *Lycium barbarum* polysaccharides (LBP), which are comprised of 5–8% in dried fruits, had the most biological activity in goji berry [13]. LBP mainly contained nine monosaccharides, namely: xylose, rhamnose, glucose, mannose, galactose, arabinose, fructose, fucose, and ribose, and had various activities such as antioxidation, immunomodulation, antitumor, and neuroprotection [14].

Capsaicin (CAP), a member of the vanilloid family with the molecular weight of 305.4 kDa, is a naturally occurring alkaloid and the primary capsaicinoid in chili pepper [15,16]. CAP was known for its analgesic effect via binding to transient receptor potential cation channel V1 (TRPV1), and also had antioxidative, anti-inflammatory, antitumor, and gastroprotective effects [17]. LBP was well known for its immunomodulation, and CAP was also well recognized for its anti-analgesic effects, suggesting that the mixture of both substances was potentially able to exert a synergistic effect against ulcerative colitis on the regulations of inflammatory responses and pain relief. We hypothesized that LBP and/or CAP could have positive effects against colitis symptoms, therefore, our study used dextran sulfate sodium (DSS) to induce colitis in rats, investigated the effects of LBP and/or CAP against colitis symptoms, and determined whether the combination of LBP and CAP could have further synergistic effects on oxidative stress, inflammatory cytokines, and the expression of pain signaling proteins.

## 2. Results

### 2.1. Effects of LBP and CAP on Body Weight and Colitis Symptoms

There were no significant differences in body weight among the groups at weeks 0 and 1 (Table 1). Food intake among the groups was not altered at weeks 1 and 4. After DSS induction at week 3, body weight in the ulcerative colitis (U) group tended to be decreased but did not reach a significant difference compared to the control group. The mixed LBP and CAP (M) group showed a significant increase in body weight compared to the U group (*p* < 0.05). Surprisingly, a significant decrease in body weight in the U group was profound during the recovery week at week 4 compared to the control group (*p* < 0.05). The M group remained significantly higher in body weight compared to the U group. The disease activity index (DAI) score of the U group was dramatically increased to the peak on the last day (day 6) of DSS induction (Figure 1). The CAP and mixture-treated groups showed significant decreases in DAI scores compared to the U group on day 6; however, the LBP treated group had no apparent changes in DAI score compared to the U group.

### 2.2. Effects of LBP and CAP on Colon Length and Weight

The ulcerative colitis group had significantly shrunken colon length compared to the control group (Figure 2A), indicating a successful induction of colitis symptoms by providing 5% DSS in drinking water to rats. The treatment with LBP or mixture of both recovered colon length compared to the U group (*p* < 0.05), but a similar result was not found in the CAP treated group. Rats with DSS-induced colitis had a significant increase in colon weight compared to those in the control group (*p* < 0.05) (Figure 2B). Unlike the results of the colon length, a significant reduction in colon weight was only observed in the mixed LBP and CAP group (*p* < 0.05). The colon weight/length ratio was correlated to the severity of colitis, and a significant increase was found in the U group compared to the control group (*p* < 0.05) (Figure 2C). Only the treatment with a mixture of both showed a significant decrease in colon weight/length ratio compared to the U group (*p* < 0.05).

### 2.3. Effects of LBP and CAP on Histopathological Changes

Histological analysis for colonic damage in rats was examined and scored by H&E stain. The colon stained by H&E in the control group did not have inflammation and necrosis and had normal morphology in the gross structure, goblet cells, crypts, and mucosa (Figure 3A). Necrosis, mucosal edema, destruction of crypts, and infiltration of inflammatory cells were observed in UC rats induced by 5% DSS, leading to a significant increase in histological score. Although rats supplemented with LBP and/or CAP had lesser histological damage, but the results demonstrated that such intervention did not achieve significant differences compared to the U group (Figure 3B).

### 2.4. Effects of LBP and CAP on Antioxidative Enzymes and Oxidative Marker

Serum superoxide dismutase (SOD) (Figure 4A) and catalase (CAT) activities (Figure 4B) were significantly decreased, and serum malondialdehydes (MDA) levels (Figure 4C) were elevated in colitis rats induced by 5% DSS (*p* < 0.05). Serum SOD activity was increased in rats treated with CAP compared to that in UC rats (*p* < 0.05), and treatment with a mixture of both also showed a similar tendency (*p* = 0.06). Although such a result was not found in LBP treatment, increased CAT activity was shown in rats treated with LBP or CAP (*p* < 0.05). The improvement of serum MDA levels was only observed in LBP treatment (*p* < 0.05).

### 2.5. Effects of LBP and CAP on Inflammatory Cytokines and *Cyclooxygenase-2* (COX-2) Protein

The UC rats significantly elevated the proinflammatory cytokines tumor necrosis factor-α (TNF-α) levels in the colon (Figure 5A) and interleukin-6 (IL-6) levels in serum (Figure 5B) compared to the control rats (*p* < 0.05). A significant reduction in colonic TNF-α levels was found in rats treated with LBP or mixture. All treatment groups significantly decreased serum IL-6 levels (*p* < 0.05). Although colonic anti-inflammatory cytokine IL-10 levels were not significantly different between the colitis and the control rats, the CAP treated group significantly increased colonic IL-10 levels compared to the U group (*p* < 0.05) (Figure 5C). Colonic expression of cyclooxygenase-2 (COX-2) protein involved in inflammation was not different between the U and the control groups; however, the treated groups with LBP and/or CAP significantly decreased COX-2 protein expression compared to the U group (*p* < 0.05) (Figure 6A,D).

### 2.6. Effects of LBP and CAP on Pain Signaling Protein Expression

Protein expression of TRPV1 (Figure 6B,D) and transient receptor potential ankyrin 1 (TRPA1) (Figure 6C,D) was significantly elevated in the U group compared to the control group (*p* < 0.05). The up-regulated TRPV1 and TRPA1 proteins were significantly suppressed by LBP or CAP treatment (*p* < 0.05); however, treatment with a mixture of both LBP and CAP only significantly decreased TRPA1 protein (*p* < 0.05).

## 3. Discussion

Oral administration of DSS via drinking water was commonly used in the animal model to induce colitis, and DSS only exerted toxicity toward intestinal epithelial cells, disrupted the epithelial layer and intestinal wall as well as increased colon permeability and chances for the infiltration of intestinal bacteria to colon tissue, which was different from the action of oxazolone or 2,4,6-trinitrobenzenesulfonic acid for direct stimulation of inflammation [18]. In addition, the symptoms of DSS-induced colitis were highly similar to those in humans with UC symptoms such as body weight loss, diarrhea, rectal bleeding, and stool bleeding. UC was also considered as intestinal inflammation through cycles between relapse and remission [19], and DAI score was an effective tool to reflect the severity of colitis [20]. In our study, the colitis model was established by providing 5% DSS in drinking water for 6 days to induce acute inflammation, and was continuously followed-up DAI scores during the recovery week to observe if such treatment could reach or maintain the remission phase earlier. Our results demonstrated that DAI scores reached the highest value after 6 days of DSS intervention, and treatment with CAP or mixture of LBP and CAP could alleviate such symptoms because of lower DAI scores on day 6.

Disruption of colon mucosa was profound with neutrophil infiltration, which was accompanied by the massive production of ROS and an increase in lipid peroxidation MDA levels [21]. As reported, LBP had antioxidative effects and could protect against the damage of lipid peroxidation induced by free radicals, and LBP could enhance nuclear factor-erythroid 2-related factor 2 (Nrf2) protein expression in ischemia/reperfusion-induced myocardial injury model [22,23]. Similar results were also found in our study, as treatment with LBP increased CAT activity and decreased MDA levels in serum. A previous study found that CAP modulated antioxidant activity through enhancing Nrf2 protein expression and elevating serum SOD and CAT activities [24], as our study also demonstrated that CAP could increase serum CAT activity. The data suggested that rats treated with LBP or CAP could improve colitis symptoms by promoting Nrf2 protein expression to increase antioxidative capacity.

The imbalance of cytokine profiles played an important role in the progression of colitis during the inflammatory state, and excessive production of proinflammatory cytokines could lead to lesions and further the erosions of colonic mucosa [25,26]. Our study demonstrated that proinflammatory cytokines TNF-α in the colon and IL-6 levels in serum were increased after 5% DSS induction, and rats treated with LBP or CAP decreased TNF-α or both cytokines. Activation of nuclear factor-κB (NF-κB) signaling was observed in the colonic epithelial cells in colitis patients, which could further stimulate the formation of proinflammatory cytokines or mediators such as IL-6, IL-17, TNF-α, and COX-2 [27]. LBP was reported to have anti-inflammatory effects and decrease IL-6 generation via inhibiting NF-κB pathway [28,29]. Previous studies showed that CAP suppressed the phosphorylation of NF-κB through the activation of peroxisome proliferator-activated receptor-γ [24,30]. The anti-inflammatory cytokine IL-10 potentially had immunoregulatory activity by inhibiting the production of proinflammatory cytokines and was higher in active UC patients, which could be explained that IL-10 served as a damper during the active UC stage to regulate imbalanced cytokines [25,31]. Though there was no significant difference in colonic IL-10 levels between the U and the control groups, suggesting that the colitis rats may be moving toward the remission state after a week of recovery. Interestingly, a significant elevation of colonic IL-10 levels was found in rats treated with CAP, and several studies also reported that capsaicin treatment could enhance IL-10 production [32,33], supposedly by stimulating CD11b^hi^F4/80^+^ macrophages [34].

The inflammatory mediator COX-2 was well known to secrete prostaglandin E2, which could cause the inflammatory responses such as redness, swelling, heat, pain, and the infiltration of inflammatory cells [35], and was correlated to colitis scores [36]; however, protein expression of COX-2 in colon did not appear to be significantly higher in our study. Our results could be explained that colitis symptoms may reach the remission period a week after the induction of acute inflammation, and colonic expression of COX-2 was decreased when colon tissue was collected for analysis. The intervention with LBP and/or CAP exerted inhibitory effects on COX-2 protein expression, suggesting the anti-inflammatory effects of LBP and CAP. Additionally, LBP or CAP could act as prebiotics to modulate intestinal microbiota and regulate colonic inflammation [37].

TRPV1 and TRPA1 co-expressing in intestinal sensory neurons played a key role in visceral sensation and pain, and increased expression of TRPV1 and TRPA1 in the intestine was found during inflammatory state [38]. The activation of TRPV1 in sensory neurons could release neuropeptides calcitonin gene-related peptide and substance P to result in vasodilatation, plasma extravasation, leukocyte migration, and increased proinflammatory cytokines [38]. Few studies showed the activation of TRPA1 by inflammatory lipid peroxides and the inhibition of colonic inflammation and calcitonin gene-related peptide release by the antagonists of TRPV1 or TRPA1 in UC animal models, indicating that the suppression of TRPV1 or TRPA1 activation could ameliorate intestinal inflammation and relieve pain symptoms [39,40,41,42]. Therefore, pain control by suppressing TRPV1 and TRPA1 proteins could be a promising target in colitis management [42]. In our study, increased protein expression of TRPV1 and TRPA1 in the U group could be reversed by treatment with LBP or CAP, indicating that LBP or CAP may relieve pain via inhibiting protein expression of TRPV1 and TRPA1.

Both LBP and CAP exerted anti-colitis effects, but each substance showed unique effects on different markers. Serum MDA levels were only decreased by LBP because goji berry was not only enriched with polysaccharides but also consisted of other components such as carotenoids and flavonoids which potentially contributed to protective effects against oxidative stress [43], and exhibited better antioxidative function compared to CAP. While CAP demonstrated better action in anti-inflammatory effects on up-regulating IL-10 production. Although LBP or CAP had promising effects on either antioxidation or anti-inflammation, combined LBP and CAP with a half dosage did not exert synergistic effects against colitis.

## 4. Materials and Methods

### 4.1. Chemicals

DSS (TdB Labs, Uppsala, Sweden), a molecular weight of 40 kDa, was purchased from BioCommander International Co., Ltd. (Taipei, Taiwan). LBP (*L. barbarum* polysaccharides M-5000) was bought from Fengyang Biomedical Co., Ltd. (Taichung, Taiwan). The preparation of LBP was extracted by water extraction, and 50% polysaccharides were obtained in LBP extract. CAP was bought from iHerb (Irvine, CA, USA), and CAP extract contained 85% total capsaicin which was determined by HPLC method. Superoxide dismutase (SOD) (706002), catalase (CAT) (707002) and malondialdehyde (MDA) (10009055) assay kits were purchased from Cayman Chemical, Inc. (Ann Arbor, MI, USA). The ELISA kits for TNF-α (438204) were bought from BioLegend (San Diego, CA, USA), for IL-6 (OKCD01310) was from Aviva Systems Biology Corp. (San Diego, CA, USA), and for IL-10 (DY522-05) was from R&D Systems, Inc. (Minneapolis, MN, USA). Bicinchoninic acid (BCA) protein assay kit (23227) was purchased from Thermo Fisher Scientific Inc. (Waltham, MA, USA). Rabbit polyclonal antibody for COX-2 (AB15191) was obtained from Abcam Plc (Cambridge, UK). Rabbit polyclonal antibodies for TRPV1 (NB100-1617) and TRPA1(NB110-40763) were purchased from Novus Biologicals LLC (Centennial, CO, USA). Horseradish peroxidase (HRP)-conjugated donkey anti-rabbit IgG (406401) was obtained from BioLegend. Reagents for Western Lightning Plus-Enhanced Chemiluminescence (ECL) were bought from PerkinElmer, Inc. (Waltham, MA, USA).

### 4.2. Animals

Male 7-week old Sprague-Dawley rats (200–250 g) were purchased from BioLASCO Taiwan Co., Ltd. (Taipei, Taiwan). Rats were kept at the Laboratory Animal Center of Taipei Medical University. Animals were housed in standard photoperiod conditions (12 h light/12 h dark), at 22 ± 2 °C and relative humidity of 65 ± 5%. Animal use and care were approved by the Institutional Animal Care and Use Committee of Taipei Medical University (LAC-2018-0344).

### 4.3. Treatments

Rats were adapted at the Laboratory Animal Center for one week, and then divided into five groups: control, DSS-induced UC (5% DSS, U), UC treated with 100 mg LBP/kg bw (L), UC treated with 12 mg CAP/kg bw (C), and UC treated with a mixture of 50 mg LBP/kg bw and 6 mg CAP/kg bw (M) groups. Rats were treated with LBP and/or CAP via gavage from weeks 1 to 4, and 5% DSS was provided for 6 days during week 3 in drinking water. The DAI score was presented as an average score of weight loss percentage, stool consistency, and fecal occult blood (Appendix A) [44]. The score was recorded daily during induction (week 3) and recovery period (week 4). After a 4-week experimental period, rats were anesthetized and sacrificed, colon and blood samples were collected and stored at −80 °C for analysis.

### 4.4. Histological Evaluation of Colonic Ulceration

The colon tissue segment was fixed in 10% formalin buffered saline for 24 h, then embedded in paraffin, and stained with hematoxylin and eosin (H&E) for histopathological evaluation. The paraffin-embedded section was performed by the National Animal Center, and the intestinal injury was determined by the veterinarian according to histopathological grading of colitis, score was assessed as follows: abnormality of mucosal architecture, the extent of inflammation, erosion or ulceration, epithelial regeneration, and percent of involvement (Appendix A).

### 4.5. Measurement of Serum SOD, CAT, and MDA

Serum antioxidant activity and serum lipid peroxidation were assessed colorimetrically using SOD, CAT, and MDA commercial kits, and all assays were performed by following the manufacturer’s instructions. Serum SOD activity was determined at 440–460 nm, and CAT activity was assessed at 540 nm. Serum MDA levels were evaluated at 530 nm.

### 4.6. Measurement of Proinflammatory and Anti-Inflammatory Cytokines

The colonic samples were homogenized in homogenate buffer (4 mM Tris-base, 2 mM NaCl, and 1% Triton X100, pH 7.2) containing protease inhibitors and phosphatase inhibitors. The homogenate was then centrifuged at 13,000× *g* for 20 min at 4 °C, and the supernatant was collected and stored at −80 °C for cytokine analysis. Colonic TNF-α, serum IL-6, and colonic IL-10 were measured by ELISA kits, and assays were performed according to the manufacturer’s protocols. All cytokines were measured at 450 nm.

### 4.7. Electrophoresis and Western Blot for COX-2 and Pain Signaling Proteins

Colon tissue was homogenized in RIPA buffer containing protease inhibitors and phosphatase inhibitors with 5-mm stainless steel beads using a TissueLyser II homogenizer (Qiagen, Hilden, Germany) at 30 Hz for 1 min. The homogenate was centrifuged at 13,000× *g* for 15 min at 4 °C to obtain the supernatant. Protein levels were measured by BCA protein assay kit. The proteins were separated by 10% sodium dodecyl sulfate-polyacrylamide gel electrophoresis and transferred to the polyvinylidene difluoride (PVDF) membrane. The membrane was then blocked with 3% bovine serum albumin in Tris-buffered saline for 2 h. Afterward, the membrane was probed with a primary antibody against internal control β-actin (1:7500), COX-2 (1:5000), TRPV1 (1:500), or TRPA1 (1:500) primary antibody overnight at 4 °C. The PVDF membrane was then incubated with a specific rabbit anti-rabbit secondary antibody for 1 h at room temperature. After repeated washes, protein bands were detected using ECL solution (reagent A:reagent B = 1:1) and visualized by the imaging system (UVP ChemiDoc-It 515 Imaging System Vision Works 8.18). The band scanning densitometry of each protein was quantitated by Image J (1.47v, National Institutes of Health, Bethesda, MA, USA).

### 4.8. Statistical Analysis

The results were expressed as mean ± standard error of the mean (SEM). Data were analyzed by SPSS version 23 (IBM Corp., Armonk, NY, USA). Two-way analysis of variance (ANOVA) followed by Bonferroni test was used to analyze DAI scores. Multiple comparisons between any two groups were performed using one-way ANOVA and Fisher’s least significant difference test. The statistical significance was set at *p* < 0.05.

## 5. Conclusions

Colitis rats induced by 5% DSS and supplemented with 100 mg LBP/kg bw for 4 weeks reduces serum lipid peroxidation substance MDA levels, colonic TNF-α, serum IL-6 levels, and the expression of pain signaling proteins TRPV1 and TRPA1 in the colon, and enhance serum antioxidative CAT activity. Colitis rats supplemented with 12 mg CAP/kg bw for 4 weeks decrease serum IL-6 levels and protein expression of TRPV1 and TRPA1 in the colon, and increase serum SOD and CAT activities and colonic IL-10 levels. Combined LBP (50 mg/kg bw) and CAP (12 mg/kg bw) for 4 weeks attenuate colonic TNF-α, serum IL-6 levels, and colonic protein expression of TRPA1. These results demonstrate that LBP and/or CAP can be used as a potential therapeutic agent against colitis by antioxidation, anti-inflammation, and the modulation of pain signaling proteins.

## Figures and Tables

**Figure 1 ijms-23-02423-f001:**
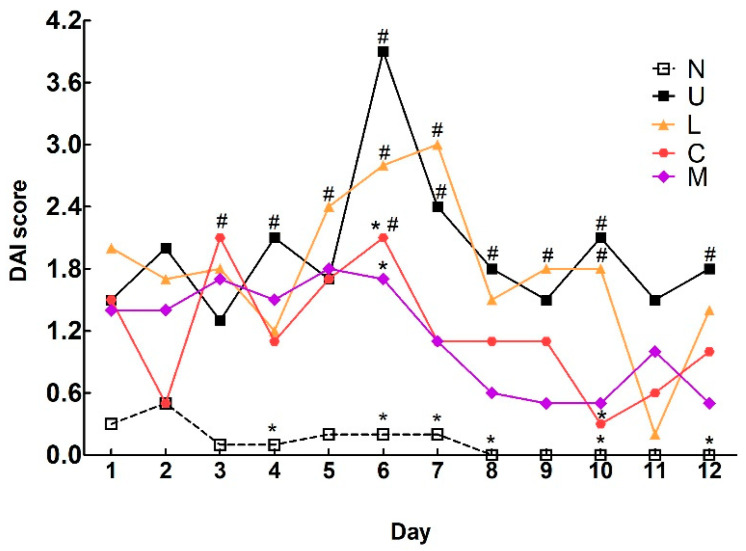
Effects of *Lycium barbarum* polysaccharides (LBP) and capsaicin (CAP) on disease activity index (DAI) scores in dextran sulfate sodium-induced colitis rats. N: control group, U: ulcerative colitis induced group, L: LBP treated group, C: CAP treated group, M: mixed LBP and CAP treated group. Data are presented as mean and analyzed by two-way ANOVA and Bonferroni test (*n* = 8). ^#^ *p* < 0.05 compared to the N group. * *p* < 0.05 compared to the U group.

**Figure 2 ijms-23-02423-f002:**
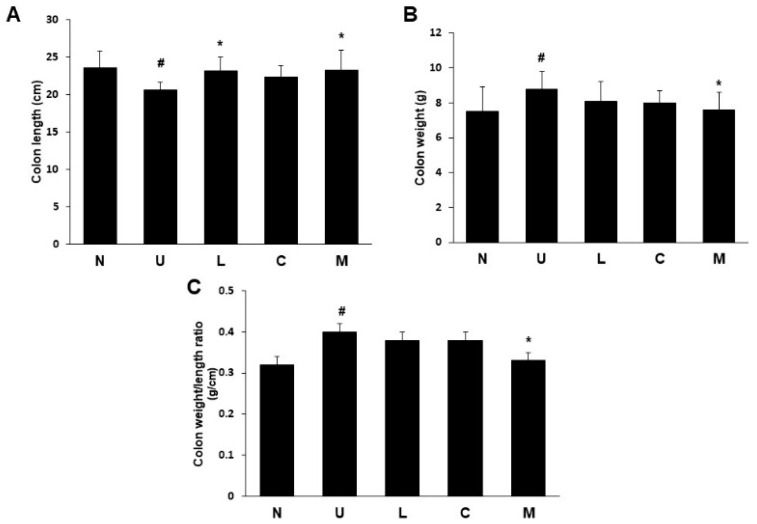
Effects of *Lycium barbarum* polysaccharides (LBP) and capsaicin (CAP) on colon length and weight in dextran sulfate sodium-induced colitis rats. (**A**) Colon length; (**B**) Colon weight; (**C**) Colon weight/length ratio. N: control group, U: ulcerative colitis induced group, L: LBP treated group, C: CAP treated group, M: mixed LBP and CAP treated group. Data are presented as mean ± SEM and analyzed by one-way ANOVA and Fisher’s least significant difference test (*n* = 8). ^#^ *p* < 0.05 compared to the N group. * *p* < 0.05 compared to the U group.

**Figure 3 ijms-23-02423-f003:**
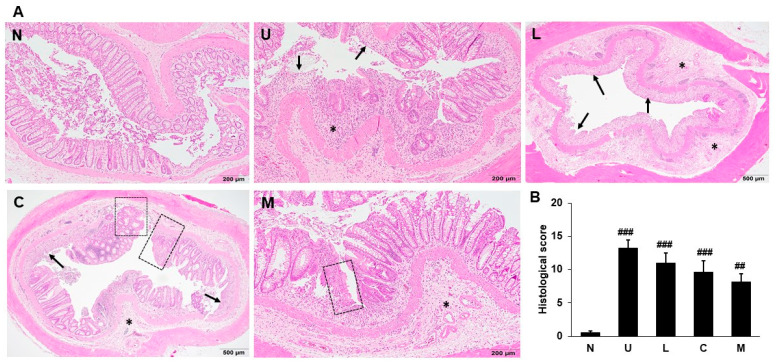
Effects of *Lycium barbarum* polysaccharides (LBP) and capsaicin (CAP) on histological changes. (**A**) Representative distal colons stained by hematoxylin and eosin; (**B**) Histological scores. N: control group, U: ulcerative colitis induced group, L: LBP treated group, C: CAP treated group, M: mixed LBP and CAP treated group. Data are presented as mean ± SEM (*n* = 5). The arrow indicates erosion or ulceration of the epithelial layer. The * symbol indicates extent of inflammation. The dotted-line square indicates regeneration of the epithelial layer. ^##^ *p* < 0.01 compared to the N group. ^###^ *p* < 0.001 compared to the N group.

**Figure 4 ijms-23-02423-f004:**
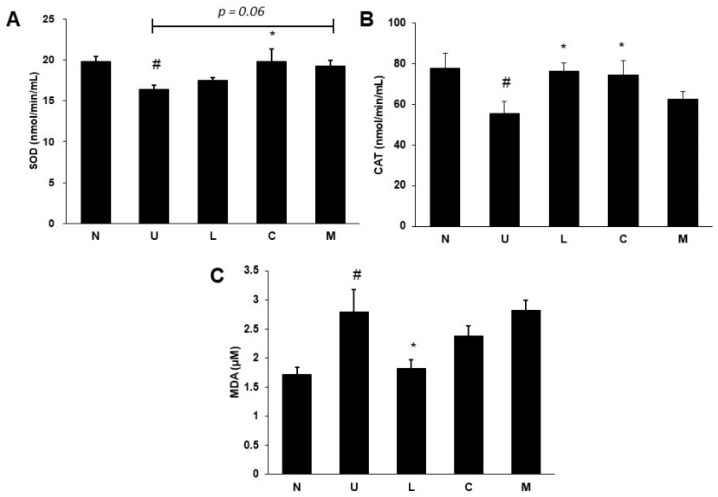
Effects of *Lycium barbarum* polysaccharides (LBP) and capsaicin (CAP) on (**A**) Superoxide dismutase (SOD) activity, (**B**) Catalase (CAT) activity, and (**C**) Malondialdehydes (MDA) levels in serum. N: control group, U: ulcerative colitis induced group, L: LBP treated group, C: CAP treated group, M: mixed LBP and CAP treated group. Data are presented as mean ± SEM and analyzed by one-way ANOVA and Fisher’s least significant difference test (*n* = 8). ^#^ *p* < 0.05 compared to the N group. * *p* < 0.05 compared to the U group.

**Figure 5 ijms-23-02423-f005:**
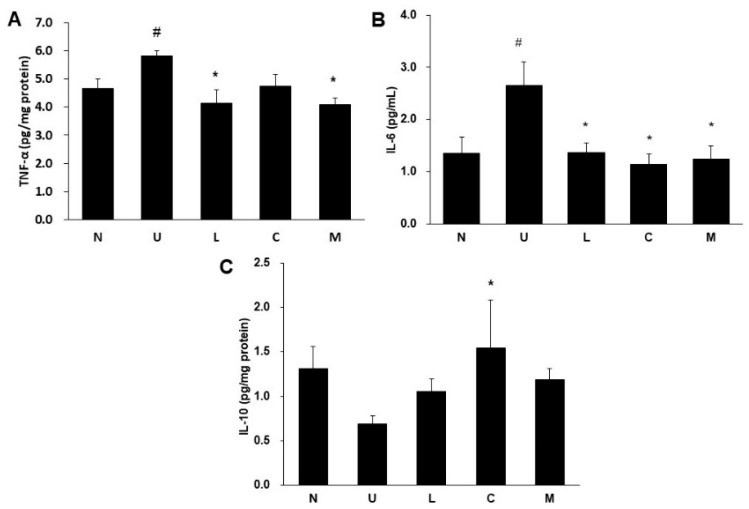
Effects of *Lycium barbarum* polysaccharides (LBP) and capsaicin (CAP) on (**A**) Colonic tumor necrosis factor-α (TNF-α), (**B**) Serum interleukin-6 (IL-6), and (**C**) Colonic interleukin-10 (IL-10). N: control group, U: ulcerative colitis induced group, L: LBP treated group, C: CAP treated group, M: mixed LBP and CAP treated group. Data are presented as mean ± SEM and analyzed by one-way ANOVA and Fisher’s least significant difference test (*n* = 8). ^#^ *p* < 0.05 compared to the N group. * *p* < 0.05 compared to the U group.

**Figure 6 ijms-23-02423-f006:**
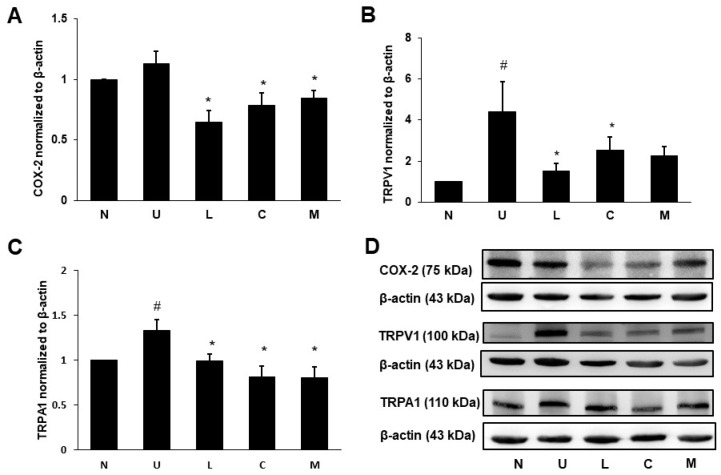
Effects of *Lycium barbarum* polysaccharides (LBP) and capsaicin (CAP) on colonic (**A**) Cyclooxygenase-2 (COX-2), (**B**) Transient receptor potential cation channel V1 (TRPV1), and (**C**) Transient receptor potential ankyrin 1 (TRPA1) protein expression. (**D**) Representatives of Western blot for COX-2, TRPV1, TRPA1, and β-actin. N: control group, U: ulcerative colitis induced group, L: LBP treated group, C: CAP treated group, M: mixed LBP and CAP treated group. Data are presented as mean ± SEM and analyzed by one-way ANOVA and Fisher’s least significant difference test (*n* = 8). ^#^ *p* < 0.05 compared to the N group. * *p* < 0.05 compared to the U group.

**Table 1 ijms-23-02423-t001:** Body weight and food intake in rats with or without dextran sulfate sodium-induced colitis ^1^.

	N	U	L	C	M
Body weight (g)					
Week 0	229.4 ± 5.9	226.3 ± 4.3	231.2 ± 4.0	229.9 ± 5.6	234.7 ± 5.6
Week 1	272.6 ± 9.1	267.6 ± 7.9	277.7 ± 9.5	274.5 ± 8.2	277.2 ± 9.4
Week 3	351.6 ± 20.9	336.3 ± 17.5	350.0 ± 19.7	341.9 ± 14.8	353.9 ± 16.2 *
Week 4	379.4 ± 27.7	357.4 ± 22.7 ^#^	370.6 ± 22.5	362.5 ± 20.6	381.2 ± 17.6 *
Food intake (g/d)					
Week 1	27.8 ± 1.3	29.5 ± 2.2	30.1 ± 2.6	29.7 ± 1.9	29.0 ± 3.4
Week 4	28.2 ± 1.8	29.2 ± 3.3	28.8 ± 2.3	28.8 ± 2.2	27.8 ± 0.8

N: control group, U: ulcerative colitis induced group, L: *Lycium barbarum* polysaccharides (LBP) treated group, C: capsaicin (CAP) treated group, M: mixed LBP and CAP treated group. ^1^ Data are presented as mean ± SEM and analyzed by one-way ANOVA and Fisher’s least significant difference test (*n* = 8). ^#^ *p* < 0.05 compared to the N group. * *p* < 0.05 compared to the U group.

## Data Availability

Data supporting the findings of the present study are not publicly available. Data are available only upon reasonable request and with permission from the authors.

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
