# Peer review of "Lycium barbarum Polysaccharides and Capsaicin Inhibit Oxidative Stress, Inflammatory Responses, and Pain Signaling in Rats with Dextran Sulfate Sodium-Induced Colitis"

_ijms, 2022, doi:10.3390/ijms23052423_

Round 1

Reviewer 1 Report

  1. In this study, LBP are polysaccharides extracted from Lycium barbarum, and CAP is a member of the vanilloid family that commonly found in chili pepper. LBP and CAP belong to different substances and possess different active ingredients. Why did the author recognize two substances without common characteristics and mix them as supplemented materials to explore their protective effects on DSS-induced ulcerative colitis in rats? Please describe the reasons in the introduction section.

  1. This study investigated the anti-analgesic effects of both LBP and/or CAP against DSS-induced ulcerative colitis, but the importance of pain management in colitis and correlation of pain signaling with inflammation were weakly described in the introduction and/or discussion sections. Please explain these mentioned above in the text, which will greatly help to understand the article more completely.

Author Response

Comments and Suggestions for Authors

  1. In this study, LBP are polysaccharides extracted from Lycium barbarum, and CAP is a member of the vanilloid family that commonly found in chili pepper. LBP and CAP belong to different substances and possess different active ingredients. Why did the author recognize two substances without common characteristics and mix them as supplemented materials to explore their protective effects on DSS-induced ulcerative colitis in rats? Please describe the reasons in the introduction section.

Reply:  Thanks for the reviewer’s comments. We revised the text accordingly in the Introduction section. We have mentioned the characteristics of LBP and CAP on lines 64-74, and both substances would have positive effects on DSS-induced ulcerative colitis, but no study had investigated that both substances could have any additive or synergistic effect. Moreover, we have added the description for the purpose of mixing two different substances on lines 74-77 as follows: “LBP was well known for its immunomodulation, and CAP was also well recognized for its anti-analgesic effects, suggesting that the mixture of both substances was potentially able to exert a synergistic effect against ulcerative colitis on the regulations of inflammatory responses and pain relief”, and line 79-82: “investigated the effects of LBP and/or CAP against colitis symptoms, and determined whether the combination of LBP and CAP could have further synergistic effects on oxidative stress, inflammatory cytokines, and the expression of pain signaling proteins.”

  1. This study investigated the anti-analgesic effects of both LBP and/or CAP against DSS-induced ulcerative colitis, but the importance of pain management in colitis and correlation of pain signaling with inflammation were weakly described in the introduction and/or discussion sections. Please explain these mentioned above in the text, which will greatly help to understand the article more completely.

Reply:  Thank you for the suggestion. We revised the text accordingly in the Discussion section. We had mentioned the correlation of inflammation with pain signaling on lines 56-59 in the Introduction section. Additionally, we have added the description about how pain management could help the relief of colitis symptoms on lines 256-260: “Few studies showed the activation of TRPA1 by inflammatory lipid peroxides and the inhibition of colonic inflammation and calcitonin gene-related peptide release by the antagonists of TRPV1 or TRPA1 in UC animal models, indicating that the suppression of TRPV1 or TRPA1 activation could ameliorate intestinal inflammation and relieve pain symptoms”. We added 2 references (#40 and #41).

Reviewer 2 Report

Authors showed whether oral administration of LBP and (or) capsaicin ameliorated DSS-induced colitis in rats. Both really improved colitis symptom and related oxidative stress.

However, following points were missed in present manuscript.

1) Why did authors select two components combination, LBP and capsaicin, and applicate to use in this colitis models? I could not image these two components combination. There was no description in the manuscript.

2) Even though both components could ameliorate colitis symptom, how about their mechanism of anti-colitis effect?

Authors simply explained their antioxidant activity (antioxidant proteins expression ), but authors have to show detail of mechanisms, for example Nrf2 and other regulatory molecules.

3) I think LBP and capsaicin can modulate intestinal microbiota. This modulation was very important for expressing anti-colitis effect. Authors should mention this possibility in the manuscript.

4) And authors should  discuss more the data in three groups, LBP, capsaicin, and both treatment groups, what is different, what is same among the groups.

5) Line 29, page 1. This description was incorrect, not included in this experiment.

Author Response

Comments and Suggestions for Authors

Authors showed whether oral administration of LBP and (or) capsaicin ameliorated DSS-induced colitis in rats. Both really improved colitis symptom and related oxidative stress.

However, following points were missed in present manuscript.

1) Why did authors select two components combination, LBP and capsaicin, and applicate to use in this colitis models? I could not image these two components combination. There was no description in the manuscript.

Reply:  Thanks for the reviewer’s comments. We revised the text accordingly in the Introduction section. We have mentioned the characteristics of LBP and CAP on lines 64-74, and both substances would have positive effects on DSS-induced ulcerative colitis, but no study had investigated that both substances could have any additive or synergistic effect. Moreover, we have added the description for the purpose of mixing two different substances on lines 74-77 as follows: “LBP was well known for its immunomodulation, and CAP was also well recognized for its anti-analgesic effects, suggesting that the mixture of both substances was potentially able to exert a synergistic effect against ulcerative colitis on the regulations of inflammatory responses and pain relief”, and line 79-82: “investigated the effects of LBP and/or CAP against colitis symptoms, and determined whether the combination of LBP and CAP could have further synergistic effects on oxidative stress, inflammatory cytokines, and the expression of pain signaling proteins.”

2) Even though both components could ameliorate colitis symptom, how about their mechanism of anti-colitis effect?

Authors simply explained their antioxidant activity (antioxidant proteins expression), but authors have to show detail of mechanisms, for example Nrf2 and other regulatory molecules.

Reply: Thank you for the suggestion. We did not further analyze the related proteins such as Nrf2 expression, but we had added the description about how both substances may have anti-colitis effects via Nrf2 in the Discussion section on lines 211-214: “As reported, LBP had antioxidative effects and could protect against the damage of lipid peroxidation induced by free radicals, and LBP could enhance nuclear factor-erythroid 2-related factor 2 (Nrf2) protein expression in ischemia/reperfusion-induced myocardial injury model”, on lines 217-218: “as our study also demonstrated that CAP could increase serum CAT activity”, and on lines 219-220: “LBP or CAP could improve colitis symptoms via promoting Nrf2 protein expression to increase antioxidative capacity”. We changed the reference source for #23.

3) I think LBP and capsaicin can modulate intestinal microbiota. This modulation was very important for expressing anti-colitis effect. Authors should mention this possibility in the manuscript.

Reply: Thank you for the suggestion. We have conducted another study in order to further investigate the effects of LBP and CAP on immunomodulation and intestinal microbiota against colitis, and this study was under review. We added the statement on lines 249-250: “Additionally, LBP or CAP could act as prebiotics to modulate intestinal microbiota and regulate colonic inflammation”. Additionally, we added I reference (#37).

4) And authors should discuss more the data in three groups, LBP, capsaicin, and both treatment groups, what is different, what is same among the group

Reply: Thank you for the suggestion. We have added the descriptions about how LBP, CAP, and both treatments at the end of the Discussion section on lines 265-273: “Both LBP and CAP exerted anti-colitis effects, but each substance showed unique effects on different markers. Serum MDA levels were only decreased by LBP because goji berry was not only enriched with polysaccharides but also consisted of other components such as carotenoids and flavonoids which potentially contributed to protective effects against oxidative stress [43], and exhibited better antioxidative function compared to CAP. While CAP demonstrated better action in anti-inflammatory effects on up-regulating IL-10 production. Although LBP or CAP had the promising effects on either antioxidation or anti-inflammation, combined LBP and CAP with a half dosage did not exert synergistic effects against colitis”. Additionally, we added 1 reference (#43).

5) Line 29, page 1. This description was incorrect, not included in this experiment.

Reply: Thank you for the suggestion. After clearly checking the whole text, we made a mistake in the Materials and Methods section, and corrected it in the text on lines 303-305: “control, DSS induced UC (5% DSS, U), UC treated with 100 mg LBP/kg bw (L), UC treated with 12 mg CAP/kg bw (C), and UC treated with mixture of 50 mg LBP/kg bw and 6 mg CAP/kg bw (M) groups”.

Round 2

Reviewer 2 Report

I don't think further corrections are necessary.